# Hypertrophy and Insulin Resistance of Epicardial Adipose Tissue Adipocytes: Association with the Coronary Artery Disease Severity

**DOI:** 10.3390/biomedicines9010064

**Published:** 2021-01-11

**Authors:** Natalia V. Naryzhnaya, Olga A. Koshelskaya, Irina V. Kologrivova, Olga A. Kharitonova, Vladimir V. Evtushenko, Alla A. Boshchenko

**Affiliations:** Cardiology Research Institute, Tomsk National Research Medical Center, Russian Academy of Science, 634050 Tomsk, Russia; oshel@live.ru (O.A.K.); ikologrivova@gmail.com (I.V.K.); hoa@cardio-tomsk.ru (O.A.K.); evtushenko.vladimir@gmail.com (V.V.E.); bosh@cardio-tomsk.ru (A.A.B.)

**Keywords:** epicardial adipose tissue, hypertrophy of adipocytes, CAD severity, adipokines, insulin resistance

## Abstract

Changes in the structural and functional characteristics of the epicardial adipose tissue (EAT) are recognized as one of the factors in the development of cardiometabolic diseases. However, the generally accepted quantitative assessment of the accumulation of EAT does not reflect the size of adipocyte and presence of adipocyte hypertrophy in this fat depot. Overall contribution of adipocyte hypertrophy to the development and progression of coronary atherosclerosis remains unexplored. Objective: To compare the morphological characteristics of EAT adipocyte and its sensitivity to insulin with the CAD severity, as well as to identify potential factors involved in the realization of this relationship. The present study involved 24 patients (m/f 16/8) aged 53–72 years with stable CAD, who underwent coronary artery bypass graft surgery. Adipocytes were isolated enzymatically from EAT explants obtained during the operation. The severity of CAD was assessed by calculating the Gensini score according to selective coronary angiography. Insulin resistance of EAT adipocytes was evaluated by reactivity to insulin. In patients with an average size of EAT adipocytes equal to or exceeding the median (87 μm) the percentage of hypertrophic adipocytes was twice as high as in patients in whom the average size of adipocytes was less than 87 μm. This group of patients was also characterized by the higher rate of the Gensini score, lower adiponectin levels, and more severe violation of carbohydrate metabolism. We have revealed direct nonparametric correlation between the size of EAT adipocytes and the Gensini score (r_s_ = 0.56, *p* = 0.00047). The number of hypertrophic EAT adipocytes showed a direct nonparametric correlation with the Gensini score (r_s_ = 0.6, *p* = 0.002). Inverse nonparametric correlations were found between the serum adiponectin level and size (r_s_ = −0.60, *p* = 0.001), hypertrophy of adipocytes (r_s_ = −0.67, *p* = 0.00), and Gensini score (r_s_ = −0.81, *p* = 0.00007). An inverse nonparametric correlation was found between the Gensini score and sensitivity of EAT adipocytes to insulin, estimated by the intracellular redox response (r_s_ = −0.90, *p* = 0.037) and decrease in lipolysis rate upon insulin addition (r_s_ = −0.40, *p* = 0.05). The intracellular redox response of adipocytes to insulin was directly correlated with fasting insulin and inversely with postprandial insulin. Our data indicate that the size and degree of hypertrophy of the epicardial adipocytes are related to the CAD severity. According to our results, insulin resistance of adipocytes may be considered as one of the factors mediating this relationship.

## 1. Introduction

Epicardial adipose tissue (EAT) has been widely proven to be an important cardiovascular risk factor due to its pronounced metabolic and humoral activity adversely affecting structural and functional state of the coronary arteries. There has been an association established of EAT with cardiovascular disease [1,2,3], hypertension [4], diabetic status [5], and insulin resistance [6].

However, in all the above-mentioned studies, authors evaluated only quantitative characteristics of EAT (its thickness or volume) in respect to the development of the cardiometabolic diseases, while the potential interconnection between morphological parameters of EAT adipocytes and coronary atherosclerosis severity was not studied. In a limited number of studies, it was shown that neither the body mass index nor the EAT depot volume are related to the size of epicardial adipocytes [7]. Differences in the size of EAT adipocytes in individuals with and without CAD (coronary artery disease) are also reported [8]. However, there is no information about possible association of hypertrophy of EAT adipocytes with the severity of CAD. It remains unclear whether the atherogenic effect of EAT is the result of its paracrine or systemic effects.

According to the results of the clinical trials, mechanisms mediating the relationship between the accumulation of EAT and the presence and severity of CAD include impaired balance of the adipokines expressed by this fat depot [9], activation of the local and systemic inflammation [10], and the development of adipose tissue fibrosis [3]. Meanwhile, the nature of factors that realize the atherogenic effects of the hypertrophied adipocytes of EAT was investigated in just a very few studies [7,8].

Insulin resistance is a well-known pathogenic factor in the development of metabolic syndrome and cardiovascular disease [11]. Excessive accumulation of EAT in insulin-resistant patients has been demonstrated [5,6]. In epicardial adipocytes, even under physiological conditions, insulin-dependent glucose uptake and anti-lipolytic function of insulin are reduced compared to adipocytes of the subcutaneous fat depot, which is associated with the need to maintain high lipolysis activity [12]. Some publications report an even more pronounced decrease in insulin sensitivity of the epicardial adipocyte pool in patients with type 2 diabetes [13,14] and low glucose transporter GLUT4 expression on epicardial adipocytes from CAD patients [15], but there is no information on the possible connection of this process with the severity of coronary atherosclerosis.

The purpose of this study: To compare the morphological characteristics of EAT adipocyte and its sensitivity to insulin with the severity of CAD in patients undergoing coronary artery bypass graft surgery, and to identify potential factors mediating this interconnection.

Hypothesis: Morphological and functional characteristics of the epicardial adipose tissue adipocytes are interrelated with the severity of coronary artery disease (CAD). We assumed that adipocyte’s hypertrophy and its level of insulin resistance are independently related to the severity of coronary atherosclerosis.

## 2. Experimental Section

The present pilot study was performed at Cardiology Research Institute, Tomsk National Research Medical Center, Russian Academy of Science, Tomsk, Russian Federation. The study’s protocol was approved by the local ethics committee, protocol nr. 146 from 16 July 2016.

### 2.1. Study Participants and Clinical Characteristics of Patients

#### Clinical Characteristics of Patients

Twenty-four patients with stable CAD, who underwent coronary artery bypass graft surgery, comprising 16 men and 8 women aged 53–72 years, were included in the study. All subjects gave their written informed consent before being enrolled in the study.

The exclusion criteria were: Age above 75 years; presence of acute atherosclerotic complications over the past 6 months; presence of concomitant diseases, including cancer, infections, chronic obstructive pulmonary disease, mental disorders, connective tissue diseases, renal insufficiency, and liver dysfunction. Hypertension was defined as a systolic blood pressure of ≥140 mmHg and/or diastolic blood pressure of ≥90 mmHg, or as the current use of antihypertensive medication. Diabetes was defined as HbA1c concentration ≥6.5% or fasting plasma glucose level >7 mM, or the current use of antidiabetic medication.

All patients received optimal therapy. The proportion of smokers and patients with metabolic syndrome was high. The clinical characteristics of patients are presented in Table 1.

All patients underwent selective coronary angiography on a Artis one angiographic complex and Digitron-3NAC computer system (Siemens Shenzhen Magnetic Resonance Ltd., Shenzhen, China). The severity of CAD was assessed by the value of the Gensini score [16]. In the blood serum, the content of leptin (Mediagnost, Reutlingen, Germany), adiponectin (Assaypro, St. Charles, MO, USA) and insulin (AccuBind kits, Diagnostic System Laboratories, Lake Forest, California, USA) were determined by enzyme-linked immunosorbent assay (ELISA). The level of glucose was detected by hexokinase assay (EKF diagnostic, Leipzig, Germany). Enzyme colorimetric method was used to estimate serum concentration of total cholesterol, triacylglycerol, high-density lipoprotein (HDL) cholesterol (Diakon, Pushchino, Russia). Concentration of low-density lipoprotein (LDL) cholesterol was calculated using formula [LDL] = [Total cholesterol] − [Triacylglycerol (TG)] − [HDL]. Atherogenic index was calculated using formula [AI] = ([Total cholesterol] − [HDL])/[HDL].

Anthropometric measurements were performed to assess total obesity according to the level of body mass index (BMI) and abdominal obesity according to the size of the waist circumference, hip circumference, and of the waist-to-hip ratio (WHR). In 16 randomly selected patients, body composition was assessed by Bioelectrical Impedance Analysis.

EAT thickness was measured on the free wall of the right ventricle in a still image at the end diastole on the parasternal long-axis view in 3 cardiac cycles. EAT thickness was measured at the point of perpendicular orientation of the ultrasound beam on the free wall of the right ventricle, using the aortic annulus as an anatomic landmark [17,18]. The thickest point of EAT was measured in each cycle. The EAT thickness was calculated as an average value from echocardiographic views in 3 cardiac cycles.

### 2.2. Adipose Tissue Explants

The material for the study were the explants of the epicardial (EAT) adipose tissue weighing 0.2–1 g obtained during the CABG surgery. Epicardial fat tissue explants were taken from the tissue surrounding the proximal parts of the right coronary artery. In all cases, electrocoagulation and other types of thermal and wave effects on tissues were not used for biopsy. Samples were placed in M199 medium and delivered to the laboratory within 15 min.

Adipose tissue cells were isolated enzymatically, in sterile conditions (laminar box BAVp-01- “Laminar-s” −1.5, ZAO “Laminar systems”, Miass, Russia) [19]. The tissue was minced, incubated for 35–40 min at 37 °C, and underwent constant gentle stirring (10 rpm) in 5 mL of type I collagenase sterile solution (PanEco, Moscow, Russia) 1 mg/mL in Krebs-Ringer buffer (2 mM D-glucose, 135 mM NaCl, 2.2 mM CaCl_2_·2H_2_O, 1.25 mM MgSO_4_·7H_2_O, 0.45 mM KH_2_PO_4_, 2.17 mM Na_2_HPO_4_, 25 mM HEPES, 3.5% BSA, 0.2 mM adenosine). Five milliliters of Krebs-Ringer buffer (37 °C) were added to the digested tissue to neutralize collagenase. The cell suspension was filtered through a nylon filter (Falcon™ Cell strainer, pore diameter 100 μm), washed three times with 10 mL of warm Krebs-Ringer buffer (37 °C). After each was, cells were allowed to float, and wash solution was discarded. The number and size of the adipocytes obtained were counted using light microscopy (Axio Observer.Z1, Carl Zeiss Surgical GmbH, Oberkochen, Germany). Cells were stained with Hoechst 33,342 (5 μg/mL, stains nucleus of viable cells) and propidium iodide (10 μg/mL, Sigma-Aldrich, St. Louis, MO, USA, stains nucleus of dead cells) to distinguish viable cells from dead cells (Figure 1) [20]. Samples with viability lower than 95% were excluded from the study. The remaining cells’ samples did not differ significantly in the percentage of viable cells.

The median values of the size of EAT adipocytes in CAD patients were 86.9 (80.97; 89.31) μm. Adipocytes with a diameter of more than 100 μm were classified as hypertrophic. The proportion of adipocytes with size exceeding 100 μm was used as a value of adipocytes’ hypertrophy. The median of this parameter constituted 14 (9.3; 18.7) %. The whole sample was divided into two groups according to the median of EAT adipocyte size and hypertrophy: If the average adipocyte size did not exceed 87 μm and percentage of hypertrophied adipocytes was less than 14%, the patient was assigned to the group of patients with non-hypertrophic adipocytes; if the average adipocyte size exceeded 87 μm and percentage of hypertrophied adipocytes was more than 14, the patient was assigned to the group of patients with hypertrophic adipocytes. All clinical parameters were calculated to these two groups (Table 1).

The sensitivity of adipocytes to insulin was estimated by increase of the production of reactive oxygen species (insulin-dependent ROS generation) [21] and by inhibition of lipolysis in response to the insulin addition to the incubation medium [13,22]. In the first case, adipocytes in 200 μL Krebs-Ringer buffer (1.25 × 10^6^ cells/mL) were added to the two wells of a 96-well plate (500,000 cells per well) and were incubated for 30 min in the presence of 125 μM 2,3-dihydrodichlorofluorescein diacetate (DCF-DA) in a microplate reader (INFINITE 200M; Tecan, Grödig, Austria) at 37 °C for the intracellular uptake and deesterification of DCF-DA to DCF in viable adipocytes. The initial fluorescence of DCF was measured at a wavelength of λex = 500, λem = 530; 20 nM of insulin was added into one of the two wells, adipocytes were incubated for 120 min inside a microplate reader at 37 °C, and the fluorescence was measured as described above. The increase in fluorescence relative to the initial values and the increase in gain under the influence of insulin were evaluated. The study was carried out in duplicates. The cell medium after 120 min of incubation of adipocytes in the previous method was harvested and used to study the inhibition of lipolysis by insulin. The medium was degreased by the Folch reaction and glycerol was determined with an EGLY-200 kit (Gentaur).

Statistical analysis was performed using the software package “Statistica” 13.0 (StatSoft Inc., Tulsa, OK, USA). The normality of the distribution of sample data was checked by the Shapiro–Wilk test. The median and interquartile ranges of the 25th and 75th percentiles were used to describe data when data distribution differed from normal. The significance of differences between quantitative indicators in the absence of normality of data distribution was verified with the Mann–Whitney test to pair comparisons; one-way ANOVA followed by Duncan’s post-hoc test were used to multiple comparisons in independent groups. When the sample data did not have normal distribution, Spearman’s rank correlation coefficient (r_s_) was adopted. Logistic regression was used to estimate the association between presence of EAT adipocytes hypertrophy, carbohydrate metabolism, adiponectin level, and atherosclerosis severity. All statistical hypotheses were accepted according to the achieved significance level *p* < 0.05.

To correct the possible modulation of the Gensini score by the gender factor, we adjusted the data for this indicator. In addition, the adipokine content was adjusted for sex and BMI.

## 3. Results

All parameters were studied depending on gender (Table 2) and ranges of BMI values (Table 3). We revealed significant differences in terms of leptin levels and adiponectin/leptin ratio between men and women (Table 2). In addition, significant differences were found in leptin levels in patients with high degrees of obesity (Table 3). To correct these effects, the adipokine content was adjusted by sex and by BMI. There were no significant differences in the Gensini score, adipocyte size, and degree of their hypertrophy depending on gender and BMI.

The median values of the size of EAT adipocytes in CAD patients were 86.9 (80.97; 89.31) μm. The proportion of adipocytes with size exceeding 100 μm was used as a value of adipocytes’ hypertrophy. The median of this parameter constituted 14 (9.3; 18.7)% (Table 1). Thus, the medians of the EAT adipocyte size in both groups differed by 8.35 μm (*p* = 0.000037), and the proportion of hypertrophic adipocytes in group of the patients with hypertrophied EAT adipocytes was twice the value of this parameter in group of patients with non-hypertrophied EAT adipocytes (Figure 2, Table 1).

The level of fasting glucose in patients from the second group exceeded the values in group of patients with non-hypertrophied EAT adipocytes by 14% (*p* = 0.46, Table 1). However, the level of fasting insulin was 1.7 times lower in group of patients with hypertrophied EAT adipocytes compared to group of patients with non-hypertrophied EAT adipocytes (*p* = 0.0039, Table 1). The parameters of postprandial glucose, postprandial insulin, glycated hemoglobin. or HOMA-IR did not differ between groups of patients with hypertrophied and non-hypertrophied EAT adipocytes. We did not reveal any differences in parameters of lipid profile, namely triacylglycerols, total cholesterol and its fractions, as well as atherogenic index (Table 1).

Patients from group with hypertrophied EAT adipocytes also had higher values of body mass index, waist circumference (Table 1) and Gensini score (Table 4) values than those in group of patients with non- hypertrophied EAT adipocytes.

The results allow us to conclude that the average EAT adipocyte size equal to or exceeding 87 μm and the proportion of hypertrophied adipocytes of more than 14% are associated with a significantly greater severity of CAD. The results of the study showed that in the group of patients with hypertrophied adipocytes, the EAT serum adiponectin adjusted to gender and BMI was 1.34 times lower than in patients with non-hypertrophic adipocytes. Based on these results, we hypothesized that adiponectin, as well as indicators of carbohydrate metabolism, may be signs associated with adipocyte hypertrophy.

Using logistic regression, a mathematical model was built that made it possible to identify significant factors influencing the likelihood of adipocyte hypertrophy EAT (Table 5):P = 1/(1 + e^−z^),
z = −2.57 + 11 FGluc − 6.7 FIns − 4.05 Adip + 0.0334 GS
where P—the probability of belonging to the second group; FGluc—fasting glucose, mM; FIns—fasting insulin, μIU/mL; Adip—serum adiponectin, μg/mL, adjusted to gender and BMI; GS—Gensini score, points, adjusted to gender.

As can be seen, the greatest association of EAT adipocyte hypertrophy is observed with an increase in fasting glucose, a decrease in fasting insulin, and a decrease in adiponectin serum. The Gensini score is less associated with EAT hypertrophy (Table 5).

The results of the study showed that the size and degree of the EAT adipocytes hypertrophy positively correlated with BMI, waist, and hip circumferences (Table 6). Interestingly, the size of EAT adipocyte and the severity of its hypertrophy were characterized by positive correlations with the Gensini score (r_s_ = 0.52, *p* = 0.009 and r_s_ = 0.41, *p* = 0.044, respectively) (Table 6, Figure 3), while there were no correlations between the size or hypertrophy of epicardial adipocyte and the thickness of the EAT. None of the anthropometric measurements of obesity, excluding EAT adipocyte size, correlated with Gensini score. Direct correlations were found between the fasting glucose level and the size of EAT adipocytes as well as with the degree of adipocyte’s hypertrophy (Table 6).

Further, we attempted to detect factors that are potentially involved in the realization of the relationships between the severity of CAD, the size of EAT adipocyte, and the degree of adipocyte hypertrophy.

In patients with an average adipocyte size less than 87 μm, the serum adiponectin levels were significantly higher than in patients with average EAT adipocyte size exceeding 87 μm (Table 4).

The severity of CAD, estimated by the Gensini score, inversely correlated with adiponectin concentrations in the blood serum (Table 7). In addition, inverse correlations were found between the serum adiponectin levels and the size and degree of hypertrophy of EAT adipocyte: r_s_ = −0.53 and r_s_ = −0.59, respectively (Table 7), whereas correlations between the adipocyte’s size and EAT adipocyte hypertrophy with serum leptin were absent.

To assess the potential correlations between the CAD severity the functional characteristics of EAT adipocytes, we evaluated the degree of insulin-dependent ROS generation and insulin-dependent inhibition of lipolysis, as parameters reflecting the sensitivity of adipocytes to insulin.

ROS generation increased 1.11 (1.00; 1.33) times in response to insulin addition to EAT adipocytes’ suspension from CAD patients compared to the control sample incubated without addition of insulin (Figure 4). Moreover, we have observed inhibition of lipolysis in response to insulin (a decrease in glycerol production compared to control values) by 1.42 (1.33; 1.78) times. Inverse correlations were found between the sensitivity of the EAT adipocytes to insulin (insulin-dependent ROS generation and insulin-dependent inhibition of lipolysis) and the values of the Gensini score (Table 6).

Direct correlation was found between insulin-dependent ROS generation and fasting insulin levels in blood serum inverse correlation was found between insulin-dependent ROS generation and postprandial insulin (Table 8).

Due to the small volume of the explant and the impossibility of carry out this study in all patients, it was impossible to assess the insulin-dependent ROS production and inhibition of lipolysis by groups depending on the size of the adipocyte.

## 4. Discussion

Our results showed that the size and hypertrophy of EAT adipocytes reflects the degree of systemic obesity and has a direct relationship with the severity of CAD. We have found that the adipocyte’s size and its degree of hypertrophy is associated with the level of fasting glucose. Since the fasting level of glucose was not significantly elevated, the development of EAT adipocyte’s hypertrophy might be assumed to precede the formation of systemic metabolic impairments. Fang L. et al. have shown that manifestation of diabetes mellitus type 2, on the contrary, is associated with an increase of the % small cells in subcutaneous and omental depots of adipose tissue, but EAT adipocytes have not been studied in this work [23].

According to our data, no relationships were found between the thickness of EAT and an average adipocyte size of the EAT. This result is in accordance with the recent studies showing that the thickness of EAT is determined not only by adipocytes’ hypertrophy, but is also dependent upon an increase of the stromal component of this tissue [7]. Adipocytes hyperplasia has been also shown to play an important role in increase of the volume of this fat depot [24]. It can be assumed that absence of direct relationships between the adipocytes’ size and EAT thickness is a consequence of the different severity of hyperplasia in patients. Differences in the degree of adipocyte hyperplasia in diabetic and non-diabetic patients were also found for another visceral fat depot—omentum [23]. However, the role of the intensity of preadipocytes differentiation (hyperplasia) of EAT in the development of CAD has not yet been studied. For the first time, we showed a significant relationship between the severity of CAD, estimated by the value of the Gensini score, the average size of EAT adipocytes, and the degree of their hypertrophy. This observation suggests that not only the quantitative determination of EAT (thickness or its volume) in itself, but also the morphological characteristics of epicardial adipocytes, reflecting the severity of their hypertrophy, namely, the average adipocyte size and proportion of cells with a size exceeding 100 μm, can be of high importance in assessing the pathological association of the epicardial fat depot with the development and progression of coronary atherosclerosis, as it allows to exclude influence of accumulation of the fibrotic component of EAT.

Adipokine imbalance, adiponectin content decrease in particular, is considered as one of the factors in the formation of coronary atherosclerosis [8,25,26]. However, the literature on the relationship of adiponectin with the formation of atherosclerosis is contradictory. Thus, the established point of view on the beneficial effects of adiponectin is based on the following data: Decrease of this parameter in patients with coronary artery disease [25]; decreased expression of adiponectin gene in epicardial adipose tissue in patients with cardiovascular disease [8]; inverse correlation of adiponectin levels and atherogenic blood plasma index [26]. Polymorphism of the gene encoding adiponectin is directly associated with the development of coronary atherosclerosis [25]. However, a meta-analysis showed that in the group of patients with elevated adiponectin levels, there was no change in the frequency of adverse cardiovascular events [27]. Moreover, distinct studies have not confirmed the difference in the severity of atherosclerosis in patients with high and low adiponectin levels [28]. One of the meta-analysis did not show a connection between transcription of adipokine genes by adipocytes of EAT and the presence of cardiovascular diseases [29]. These data indicate that information about the relationship of the adipokine profile with the degree of cardiovascular risk is incomplete.

In our study, we documented the inverse relationship of the serum adiponectin levels and the severity of CAD, which supports the point of view about the negative effect of low adiponectin on the formation of coronary atherosclerosis. It was shown that the EAT thickness is inversely related to the serum adiponectin levels regardless of the presence of CAD, as it is detected both in patients with documented CAD [30], and in patients without CAD [31]. The same pattern was found in patients with metabolic syndrome [32]. Nevertheless, there also exists evidence of the absence of such a relationship [33]. To date, comprehensive information on the ratio of the size of the adipocyte and serum adiponectin was not available in the literature. In our study, inverse correlations were established between the serum adiponectin levels, the size of the EAT adipocyte, and the degree of its hypertrophy. In addition, data were obtained on the existence of an inverse relationship between the size of EAT adipocyte, concentration of serum adiponectin, and the severity of CAD.

The mechanism of anti-atherogenic effect of adiponectin, according to the literature, includes the improvement of endothelial function and monocytes-endothelium interactions; inhibition of smooth muscle cell proliferation; reduced macrophage uptake of cholesterol and suppression of foam cells’ formation [9]. In addition to the abovementioned effects of adiponectin, it can be assumed that its anti-atherogenic effect can be achieved by improving the metabolic function of adipocytes. It was found that a decrease in the production of adiponectin in EAT is associated with a violation of oxidative phosphorylation of epicardial adipocytes and is associated with the severity of coronary atherosclerosis [34]. In addition, adiponectin was found to stimulate glucose utilization and fatty acid oxidation via adenosine monophosphate activated protein kinase (AMPK) [35]. A number of experimental and clinical studies have shown an increase in insulin-dependent uptake of glucose by adipocytes under the influence of adiponectin [36,37] and the association between a decrease in adiponectin and insulin resistance [38], of which importance in the formation of cardiovascular pathology is now generally appreciated [11]. However, the question on the role of insulin resistance of epicardial fat adipocytes in the development of atherosclerosis remains poorly understood.

This study found a direct correlation between insulin resistance of EAT adipocytes and the severity of coronary atherosclerosis which is consistent with previous studies. Even more so, we demonstrated that insulin sensitivity of EAT adipocyte directly correlates with decreased fasting insulin concentration and inversely—with its postprandial production. Thus, we are the first to show that the degree of EAT adipocytes’ insulin resistance is associated with the severity of atherosclerosis and systemic imbalance of insulin production independently from the size and hypertrophy of adipocytes.

EAT is known to be characterized by the certain basic level of insulin resistance. Even under the normal conditions, the expression of GLUT4 is reduced in EAT compared to subcutaneous adipose tissue. Thus, insulin-dependent glucose uptake and anti-lipolytic function of insulin in this fat depot are reduced. Adipocytes of CAD patients are characterized by an even more pronounced decrease in GLUT4 and an increased content of retinol-binding protein-4 (RBP4), associated with the development of insulin resistance [13,39]. Epicardial adipocytes are characterized by reduced expression of genes regulating lipid metabolism, in particular lipoprotein lipase being dependent on the size of the adipocyte, which leads to stimulation of lipolysis in hypertrophic adipocytes [40]. Violation of insulin sensitivity of EAT adipocytes can lead to even greater decrease in the ability of the epicardial fat depot to accumulate fatty acids and, thereby, regulate fatty acids’ flow to the myocardium. This mechanism may cause, in particular, the absence of excessive accumulation of EAT in a certain cohort of CAD patients. It should be noted, however, that this assumption needs further proof. Indirect evidence of the important role of EAT adipocytes’ insulin resistance in the formation of cardiovascular disease is provided by the fact that pharmacological increase in the sensitivity of adipocytes to glucose, induced by the sodium glucose cotransporter-2 inhibitor (SGLT2) dapagliflozin [41], reduces the risk of adverse cardiovascular events in patients with type 2 diabetes mellitus [42].

Our study is one of the few in which the cellular mechanisms of the relationships between EAT and the severity of coronary atherosclerosis are addressed in a clinical setting. Further studies involving a comparison group and prospective observation are required for substantiating the expansion of the cluster of the metabolic syndrome and improving the stratification of cardiovascular risk in patients with visceral obesity.

Limitations of this study are its cross-sectional nature and small sample of patients, which did not allow to study potential gender differences in relationships between morphological and functional characteristics of the epicardial adipose tissue adipocytes and CAD severity. The available amount of EAT collected from several patients was insufficient to perform all experiments within a portion of the same sample. All the recruited patients continuously received statins, which are known to affect adipocyte size. We presume that this might have affected absence of associations between the parameters of lipid profile, adipokines’ concentrations and severity of atherosclerosis.

## 5. Conclusions

Morphological and functional characteristics of adipocytes from EAT are interrelated with the severity of atherosclerotic lesions of the coronary arteries. Violation of insulin sensitivity of EAT adipocytes and an imbalance in the secretion of adipokines are among the most plausible factors determining this relationship. Adipocyte’s hypertrophy and its level of insulin resistance are independently related to the severity of coronary atherosclerosis. Hypertrophic adipocytes size are associated with insulin resistance, low plasma circulating adiponectin levels and CAD severity.

## Figures and Tables

**Figure 1 biomedicines-09-00064-f001:**
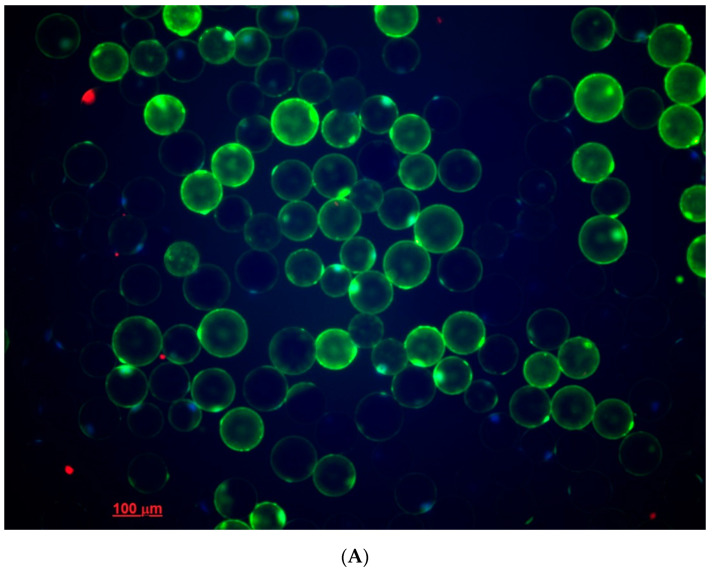
Accumulation of reactive oxygen species (ROS) and viability of adipocytes in epicardial adipose tissue (EAT) culture. (**A**) Fluorescence staining. Dyes: Green—2,3-dihydrodichlorofluorescein (ROS), red—Propidium iodid (dead cells), blue—Hoechst 33,342 (viable cells). (**B**) Corresponding light microscopy of EAT adipocytes. Magnification × 200. EAT, epicardial adipose tissue.

**Figure 2 biomedicines-09-00064-f002:**
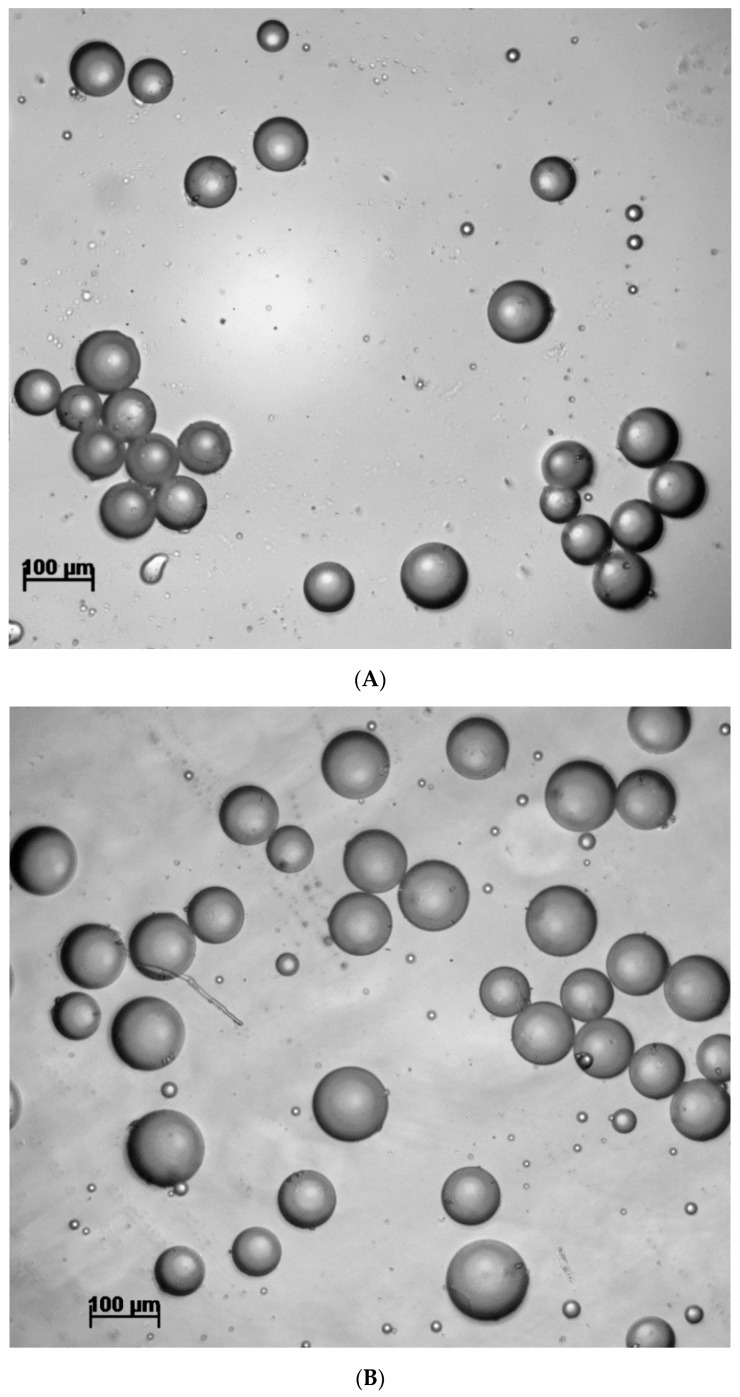
The representative snapshots of EAT adipocytes of patients from groups of patients with non-hypertrophied EAT adipocytes and with hypertrophied EAT adipocytes. (**A**) Patient from group of patients with non-hypertrophied EAT adipocytes, median of adipocytes diameter is 80.27 μm; (**B**) patient from group of patients with hypertrophied EAT adipocytes, median of adipocytes diameter is 89.88 μm. Light microscopy, magnification × 200. EAT, epicardial adipose tissue.

**Figure 3 biomedicines-09-00064-f003:**
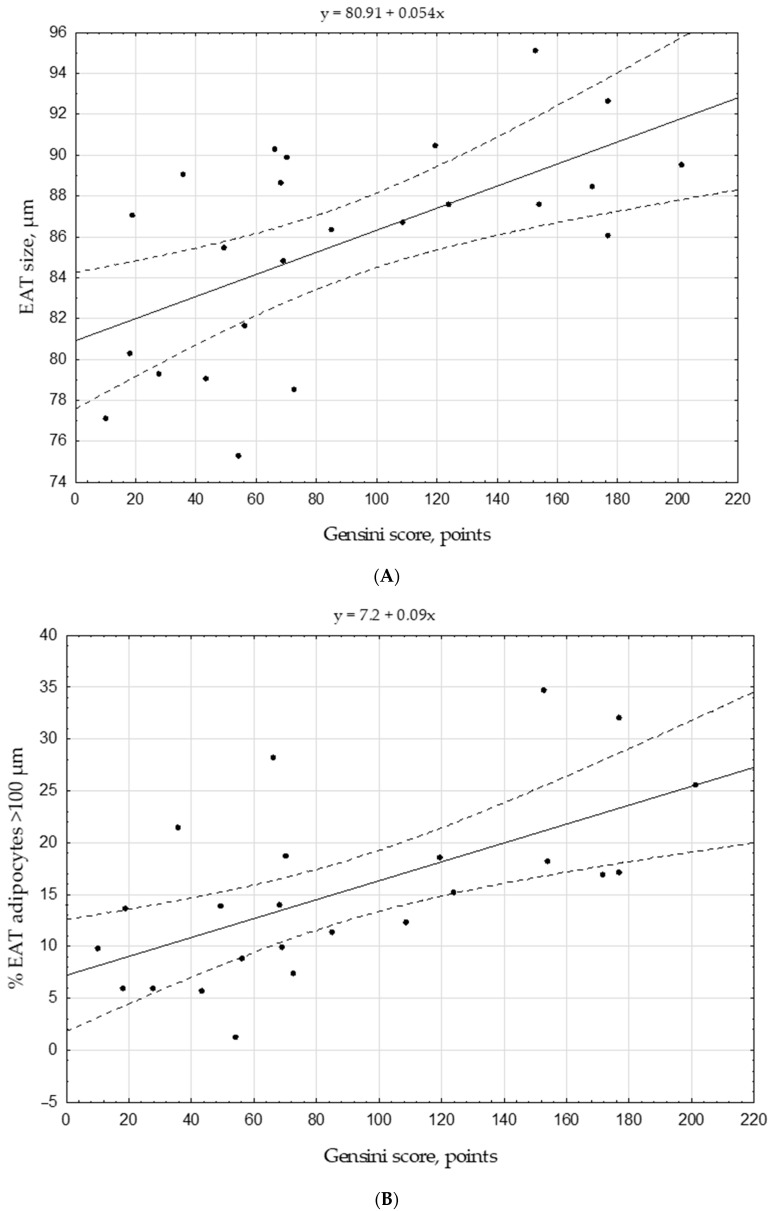
Scatter plots displaying the size of EAT adipocyte (**A**) and the proportion of hypertrophied adipocytes (**B**) related to the Gensini score. Note: EAT, epicardial adipose tissue. Gensini score was adjusted to gender.

**Figure 4 biomedicines-09-00064-f004:**
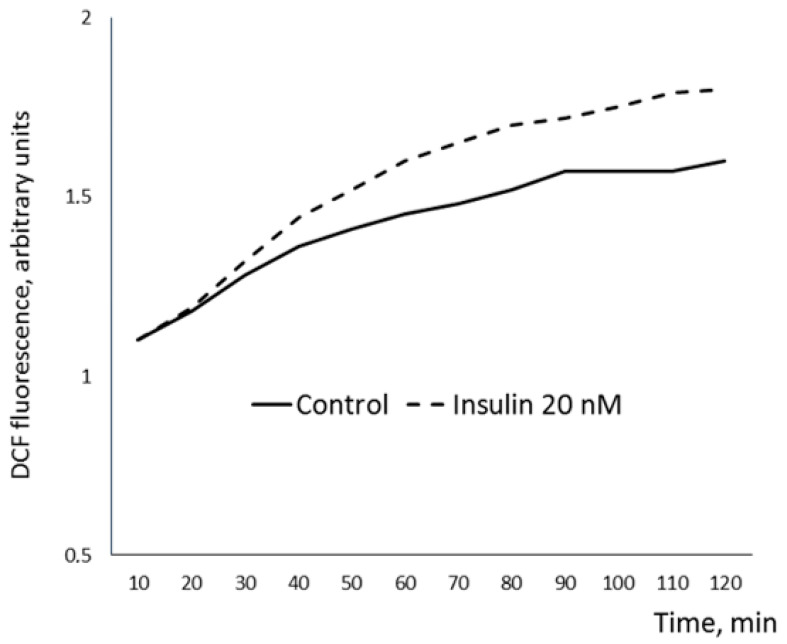
Representative dynamics of insulin-dependent ROS generation in EAT adipocytes. X axis—time, min; Y axis—2,3-dihydrodichlorofluorescein (DCF) fluorescence, arbitrary units, reference to blank. EAT, epicardial adipose tissue; ROS, reactive oxygen spices.

**Table 1 biomedicines-09-00064-t001:** Clinical characteristics of patients depending on the hypertrophy of adipocytes of epicardial adipose tissue.

Parameters	Total Sample (*n* = 24)	Patients with Non-Hypertrophied EAT Adipocyte, (*n* = 12)	Patients with Hypertrophied EAT Adipocyte, (*n* = 12)	*p*
Gender (m/f)	16/8	7/5	9/3	0.1
Age, years	62 (53–72)	62 (53–71)	59 (55–72)	0.64
History of myocardial infarction, *n* (%)	9 (37.5%)	5 (42%)	4 (33%)	0.1
Hypertension, *n* (%)	24 (100%)	12 (100%)	12 (100%)	1
Diabetes mellitus, *n* (%)	7 (29.2%)	3 (25%)	4 (33%)	0.9
Duration of hypertension, years	15 (10; 21)	15 (10; 20)	20 (12; 23)	0.3
Gensini score, points *	70 (28; 99)	32 (25.75; 78)	82 (52.5; 140.75)	0.024
Duration of CAD, years	2 (1; 7)	2 (1; 11)	5 (2; 6)	0.29
Systolic blood pressure, mmHg	136 (127; 142)	130 (123; 141)	140(135; 144)	0.38
Diastolic blood pressure, mmHg	80 (73; 85)	77.5 (70; 83)	81 (74; 86)	0.49
Smoking, *n* (%)	11 (46%)	6 (50%)	5 (42%)	0.1
Obesity, *n* (%)	12 (50%)	4 (33%)	8 (66%)	0.1
BMI, kg/m^2^	30 (27; 31)	28.1 (25.5; 30.3)	31.2 (29.8; 35.4)	0.028
Waist circumference, cm	104 (98; 110)	100 (99; 105)	109 (103; 117)	0.0083
Waist-to-hip ratio	1 (0.93; 1.04)	1 (0.9; 1.02)	1 (0.93; 1.09)	0.21
Fat mass, kg	30.55 (26.4; 37.4)	27.7 (24.4; 36.7)	30.7 (27.5; 38.1)	0.53
Fat free mass, kg	57.6 (47.1; 61.7)	57.5 (47.1; 58.3)	57.7 (47.0; 62.3)	0.65
Skeletal muscle mass, kg	26.3 (19.7; 28.7)	24.3 (22.4; 27.2)	26.8 (18.0; 29.2)	0.82
EAT thickness, mm	4.5 (4.1; 5.4)	4.85(4.36; 5.6)	4.35 (3.88; 4.9)	0.22
EAT adipocytes size, μm	86.9 (80.97; 89.31)	80.96 (78.8; 85.75)	89.31 (88.06; 90.39	0.000037
% EAT adipocytes >100 μm	14 (9.32; 18.65)	9.32 (5.91; 11.87)	18.65 (16.08; 26.88)	0.000014
Fasting glucose, mM	5.7 (5.13; 6.13)	5.25 (5.1; 5.65)	5.98 (5.75; 7.3)	0.046
Fasting insulin, μIU/mL	5.7 (5.13; 6.13)	8.22 (6.07; 9.15)	4.95 (3.5; 5.38)	0.0039
Postprandial glucose, mM	7.1(5.7; 7.8)	7.025 (5.7; 7.7)	7.1 (5.7; 7.89)	0.96
Postprandial insulin, μIU/mL	15.45 (11.4; 21.17)	14.13 (13.34; 18.1)	16.76 (11.4; 21.17)	0.87
Glycated hemoglobin, %	6.35 (5.54; 6.92)	6.7 (5.65; 7.32)	5.91 (5.46; 6.5)	0.10
HOMA-IR	1.6 (1.15; 2.01)	1.84 (1.43; 2.29)	1.38 (0.76; 1.76)	0.088
Total cholesterol, mM	3.88 (3.25; 4.58)	3.53 (3.1; 4.28)	4.21 (3.71; 4.79)	0.23
TG, mM	1.35 (1.12; 1.58)	1.24 (0.94; 1.41)	1.44 (1.23; 1.85)	0.078
HDL, mM	1.04 (0.92; 1.21)	1.0 (0.83; 1.18)	1.06 (0.99; 1.36)	0.20
LDL, mM	2.11 (1.66; 2.55)	2.0 (1.57; 2.63)	2.25 (1.76; 2.55)	0.62
Atherogenic index	2.14 (1.5; 2.6)	2.22 (1.36; 2.9)	2.05 (1.52; 2.37)	0.58

Note: data are presented as median (Me) and interquartile range (Q_25%_; Q_75%_); *p*—significance level of differences between genders (Mann–Whitney U-test). CAD, coronary artery disease, EAT, epicardial adipose tissue; BMI, body mass index; HOMA-IR, homeostatic model assessment of insulin resistance; TG, triacylglycerols; LDL, low density lipoprotein; HDL, high density lipoprotein.

**Table 2 biomedicines-09-00064-t002:** Clinical characteristics of coronary artery disease (CAD) patients depending on gender.

Parameters	Total Sample (*n* = 24)	Men (*n* = 16)	Women (*n* = 8)	*p*
Age, years	62 (53–72)	59 (53–71)	63 (56–72)	0,21
BMI, kg/m^2^	30 (27; 31)	29 (26; 31)	31 (30; 33)	0.11
Waist circumference, cm	104 (98; 110)	107 (99; 112)	101 (95; 107)	0.35
Waist-to-hip ratio	1 (0.93; 1.04)	1.02 (0.96; 1.05)	0.93 (0.91; 1.03)	0.14
Fat mass, kg	30.55 (26.4; 37.4)	30.3 (26.4; 36.5)	37.4 (28.5; 44.5)	0.33
Fat free mass, kg	57.6 (47.1; 61.7)	58.95 (52.6; 62.05	44.2 (44.0; 51.0)	0.025
Skeletal muscle mass, kg	26.3 (19.7; 28.7)	27.0 (23.35; 29.25	16.2 (15.10; 22.45)	0.95
EAT adipocytes size, μm	86.9 (80.97; 89.31)	87.3 (82.55; 88.87)	85.75 (80.10; 91.49)	0.92
% EAT adipocytes >100 μm	14 (9.32; 18.65)	13.86 (9.84; 18.43)	15.54 (8.16; 30.13)	0.6
% EAT adipocytes <50 μm	2.08 (0.84; 3.86)	1.84 (0.99; 3.1)	3.67 (0.43; 5.09)	0.49
Gensini score, points	70 (28; 99)	71 (32; 123)	35 (28; 82)	0.31
EAT thickness, mm	4.5 (4.1; 5.4)	4.36 (4.0; 5.0)	4.98 (4.5; 5.6)	0.19
Fasting glucose, mM	5.7 (5.1; 6.1)	5.7 (5.1; 6.3)	5.8 (5.4; 6.1)	0.61
Fasting insulin, μIU/mL	5.7 (5.13; 6.13)	5.49 (3.56; 7.62)	7.03 (5.01; 11.12)	0.14
Postprandial glucose, mM	7.1(5.7; 7.8)	6.7 (5.7; 7.7)	7.5 (6.8; 7.9)	0.38
Postprandial insulin, μIU/mL	15.45 (11.4; 21.17)	14.13 (11.87; 21.17)	16.76 (11.40; 17.12)	0.95
Glycated hemoglobin, %	6.35 (5.54; 6.92)	5.8 (5.46; 6.8)	6.69 (6.11; 9.09)	0.17
HOMA-IR	1.6 (1.15; 2.01)	1.6 (0.76; 1.77)	1.7 (1.27; 3.79)	0.23
Total cholesterol, mM	3.88 (3.25; 4.58)	3.87 (3.12; 4.38)	3.94 (3.58; 4.82)	0.54
TG, mM	1.35 (1.12; 1.58)	1.32(1.14; 1.44)	1.48 (1.01; 2.18)	0.43
HDL, mM	1.04 (0.92; 1.21)	1.05 (0.96; 1.18)	1.00 (0.87; 1.31)	0.83
LDL, mM	2.11 (1.66; 2.55)	1.98 (1.62; 2.57)	2.25 (1.93; 2.54)	0.56
Atherogenic index	2.14 (1.5; 2.6)	2.03 (1.36; 2.8)	2.24 (1.67; 2.39)	0.6
Adiponectin, μg/mL	7.25 (4.85; 9.91)	7.64 (4.77; 10.07)	6.70 (4.85; 9.92)	0.73
Leptin, ng/mL	17.49 (8.63; 27.01)	11.33 (7.36; 17.49)	40.25 (23.68; 67.58)	0.0017
Adiponectin/leptin	0.46 (0.17; 0.80)	0.8 (0.37; 1.08)	0.13 (0.08; 0.20)	0.0077

Note: data are presented as median (Me) and interquartile range (Q_25%_; Q_75%_); *p*—significance level of differences between genders (Mann–Whitney U-test). CAD, coronary artery disease, EAT, epicardial adipose tissue; BMI, body mass index; HOMA-IR, homeostatic model assessment of insulin resistance; TG, triacylglycerols; LDL, low density lipoprotein; HDL, high density lipoprotein.

**Table 3 biomedicines-09-00064-t003:** Clinical characteristics of CAD patients depending on BMI.

Parameters	Total Sample (*n* = 24)	BMI < 25 (*n* = 3)	25 < BMI < 30 (*n* = 9)	30 < BMI < 35 (*n* = 8)	35 < BMI < 40 (*n* = 4)
Age, years	62 (53–72)	59 (55–71)	61 (53–67)	63 (55–71)	63 (57–72)
BMI, kg/m^2^	30 (27; 31)	24.6 (20.3; 24.7)	28.1 (27.4; 29.1) *,&,‡	31.2 (30.8; 31.4) *,#,‡	37.6 (36.1; 39.5) *,#,&
Waist circumference, cm	104 (98; 110)	98 (76; 100)	102 (97; 106)	109 (104; 110)	119 (105; 120)*
Waist-to-hip ratio	1 (0.93; 1.04)	1.02 (0.84; 1.02)	0.94 (0.9; 0.98)	1.04 (1.01; 1.08)	0.95 (0.92; 1.02)
EAT adipocytes size, μm	87 (81; 89)	87 (79; 87)	85 (79; 88)	87 (83; 89)	90 (89; 92)
% EAT adipocytes >100 μm	14 (9.3; 18.7)	12.3 (5.7; 18.2)	11.4 (5.9; 15.2)	13.9 (9.4; 17.9)	26.9 (23.5; 30.1) *,#
Gensini score, points	70 (28; 99)	110 (44; 156)	28 (24.5; 121)	69.5 (32; 73.5)	62 (34.5; 146)
EAT thickness, mm	4.5 (4.1; 5.4)	4.1 (3.5; 4.9)	4.35 (4.0; 4.4)	5.35 (4.60; 5.94)	4.65 (4.18; 4.9)
Fasting glucose, mM	5.7 (5.1; 6.1)	5.1 (4.3; 5.15)	5.3 (5.1; 6.0)	5.85 (5.55; 7.11)	6.24 (5.79; 7.6)
Fasting insulin, μIU/mL	5.7 (5.13; 6.13)	2.47 (1.85; 5.62)	7.48 (4.98; 8.6)	5.93 (4.97; 11.1)	5.71 (4.97; 6.9)
Postprandial glucose, mM	7.1(5.7; 7.8)	5.7 (4.9; 7.7)	6.9 (5.7; 7.9)	7.24 (5.58; 7.90)	7.34 (7.3; 7.37)
Postprandial insulin, μIU/mL	15.45 (11.4; 21.17)	18.1 (10.08; 20.1)	13.59 (11.06; 32.31)	15.62 (11.40; 41.67)	14.32 (11.87; 16.76)
Glycated hemoglobin, %	6.35 (5.54; 6.92)	5.65 (5.27; 7.32)	6.07 (5.59; 6.43)	6.8 (5.57; 9.09)	6.5 (5.5; 6.8)
HOMA-IR	1.6 (1.15; 2.01)	0.47 (0.42; 1.27)	1.7 (1.15; 2.03)	1.68 (1.38; 3.63)	1.59 (1.28; 2.37)
Total cholesterol, mM	3.88 (3.25; 4.58)	3.28 (3.22; 5.12)	3.76 (3.58; 4.31)	4.21 (2.68; 4.89)	3.9 (3.03; 4.42)
TG, mM	1.35 (1.12; 1.58)	1.37 (0.77; 1.45)	1.27 (1.14; 1.50)	1.43 (1.12; 2.01)	1.49 (1.19; 1.72)
HDL, mM	1.04 (0.92; 1.21)	1.09 (0.64; 1.18)	1.04 (1.01; 1.36)	0.9 (0.81; 1.26)	1.06 (1.01; 1.15)
LDL, mM	2.11 (1.66; 2.55)	1,95 (1.75; 3.36)	2.04 (1.68; 2.42)	2.13 (1.39; 2.56)	2.27 (1.92; 2.48)
Atherogenic index	2.14 (1.5; 2.6)	3,05 (1.48; 3.09)	2,12 (1.24; 2.45)	1.87 (1.37; 2.58)	2.17 (1.84; 2.25)
Adiponectin, μg/mL	7.25 (4.85; 9.91)	8.58 (6.26; 24.38)	5.73 (4.6; 11.94)	7.46 (4.85; 9.92)	6.89 (6.14; 7.64)
Leptin, ng/mL	17.49 (8.63; 27.01)	5.96 (1.52; 6.40)	11.64 (8.63; 23.65)	27.83 (16.74; 48.52)	48.23 (20.72; 75.74) *,#
Adiponectin/leptin	0.46 (0.17; 0.80)	1.34 (1.05; 1.04)	0.56 (0.25; 0.8)	0.18 (0.11; 0.46)	0.23 (0.08; 0.37)

Note: data are presented as median (Me) and interquartile range (Q_25%_; Q_75%_); *—significance level of differences to BMI < 25; #—significance level of differences to 25 < BMI < 30; &—significance level of differences to 30 < BMI < 35; ‡—significance level of differences to 35 < BMI < 40; *p* < 0.05, one-way ANOVA followed by Duncan’s post-hoc test. CAD, coronary artery disease, EAT, epicardial adipose tissue; BMI, body mass index; HOMA-IR, homeostatic model assessment of insulin resistance; TG, triacylglycerols; LDL, low density lipoprotein; HDL, high density lipoprotein.

**Table 4 biomedicines-09-00064-t004:** Coronary artery disease severity and adipokine’s levels adjusted to gender and BMI depending on the hypertrophy of adipocytes of epicardial adipose tissue.

Parameters	Total Sample (*n* = 24)	Patients with Non- Hypertrophied EAT Adipocyte (1) (*n* = 12)	Patients with Hypertrophied EAT Adipocyte (2) (*n* = 12)	*p*
* Gensini score, points	69.5 (46.3; 138.3)	55.28 (35.50; 78.58)	121.52 (67.18; 162.69)	0.05
# Serum adiponectin, μg/mL	7.31 (5.73; 9.59)	8.29 (7.31; 14.81)	6.15 (5.40; 7.46)	0.039
# Serum leptin, ng/mL	17.47 (10.73; 22.35)	17.47 (12.71; 20.78)	18.11 (9.25; 40.38)	0.69
# Serum adiponectin/leptin	0.48 (0.25; 0.752)	0.57 (0.48; 0.75)	0.24 (0.14; 0.65)	0.06
Fasting glucose, mM	5.7 (5.13; 6.13)	5.25 (5.1; 5.65)	5.98 (5.75; 7.3)	0.046
Fasting insulin, μIU/mL	5.51 (4.78; 8.22)	8.22 (6.07; 9.15)	4.95 (3.5; 5.38)	0.0039

Note: data are presented as median Me and interquartile range (Q_25%_; Q_75%_); *—data were adjusted to gender; #—data were adjusted to gender and BMI; *p*—significance level of differences between the patients with non-hypertrophied EAT adipocyte and patients with hypertrophied EAT adipocyte (Mann–Whitney U-test). BMI, body mass index, EAT, epicardial adipose tissue.

**Table 5 biomedicines-09-00064-t005:** Logistic regression results.

Parameters	Regression Coefficient	*p*
Fasting glucose, mM	11	<0.0001
Fasting insulin, μIU/mL	−6.7	<0.0001
Serum adiponectin, μg/mL	−4.05	<0.0001
Gensini score, points	0.0334	0.048

Note: Serum adiponectin level adjusted to gender and BMI; Gensini score adjusted to gender.

**Table 6 biomedicines-09-00064-t006:** Correlation between the size of EAT adipocyte and the degree of its hypertrophy with the measurements of obesity, carbohydrates metabolism, and Gensini score.

	EAT Adipocytes Size		EAT Adipocytes Hypertrophy	
Parameters	r_s_	p	r_s_	p
BMI, kg/m^2^	0.45	0.028	0.59	0.002
Waist circumference, cm	0.38	0.063	0.34	0.11
Hip circumference, cm	0.41	0.049	0.51	0.01
Waist-to-hip ratio	0.16	0.46	0.011	0.96
EAT thickness, mm	−0.33	0.11	−0.30	0.16
* Gensini score, points	0.56	0.00047	0.6	0.002
CAD duration, years	0.29	0.16	0.25	0.23
Fasting glucose, mM	0.43	0.034	0.43	0.037
Fasting insulin, μIU/mL	−0.37	0.076	−0.23	0.28
Postprandial glucose, mM	0.10	0.67	0.08	0.73
Postprandial insulin, μIU/mL	−0.25	0.32	−0.12	0.63
Glycated hemoglobin, %	−0.13	0.59	−0.07	0.78
HOMA-IR	−0.21	0.38	−0.07	0.79
Total cholesterol, mM	0.17	0.42	0.17	0.43
TG, mM	0.34	0.11	0.3	0.15
HDL, mM	0.20	0.35	0.18	0.40
LDL, mM	0.11	0.61	0.12	0.57
Atherogenic index	−0.05	0.83	−0.02	0.93

*—data were adjusted to gender; r_s_, Spearman rank correlation coefficient; CAD, coronary artery disease; EAT, epicardial adipose tissue; BMI, body mass index; HOMA-IR, homeostatic model assessment of insulin resistance; TG, triacylglycerols; LDL, low density lipoprotein; HDL, high density lipoprotein.

**Table 7 biomedicines-09-00064-t007:** Correlation between serum adiponectin, leptin, EAT adipocyte size, EAT adipocyte hypertrophy, and Gensini score.

	EAT Adipocyte Size		EAT Adipocyte Hypertrophy		Gensini Score	
Parameters	r_s_	p	r_s_	p	r_s_	p
Serum adiponectin	−0.60	0.01	−0.67	0.0029	−0.81	0.00007
Serum leptin	0.08	0.75	0.027	0.91	0.30	0.23
Serum adiponectin/leptin	−0.50	0.04	−0.48	0.047	−0.65	0.0044

Note: data are presented as median (Me) and interquartile range (Q_25%_; Q_75%_); r_s_—Spearman rank correlation coefficient; adipokines’ levels were adjusted to gender and body mass index; Gensini score was adjusted to gender; EAT, epicardial adipose tissue.

**Table 8 biomedicines-09-00064-t008:** The correlation between insulin sensitivity and insulin resistance of EAT adipocyte with Gensini score or postprandial insulin.

Parameter	r_s_	p
Insulin-dependent ROS generation/Gensini score	−0.90	0.037
Insulin-dependent inhibition of lipolysis/Gensini score	−0.40	0.051
Insulin-dependent ROS generation/Fasting insulin	0.90	0.037
Insulin-dependent ROS generation/Postprandial insulin	−0.90	0.037

Note: ROS, reactive oxygen spices; r_s_ Spearman rank correlation coefficient; Gensini score was adjusted to gender. EAT, epicardial adipose tissue; ROS, reactive oxygen spices.

## Data Availability

The data presented in this study are available on request from the corresponding author.

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
