# Peer review of "Hypertrophy and Insulin Resistance of Epicardial Adipose Tissue Adipocytes: Association with the Coronary Artery Disease Severity"

_biomedicines, 2021, doi:10.3390/biomedicines9010064_

Round 1
Reviewer 1 Report
The authors have included some data corrected by gender and BMI. However, they did not perform glucose uptake assays.
Some points should be improved:
a) Table 1: EAT thickness and Gensini score should be included.
b)Figure 1: Control and Insulin treatment should be included in the fluorescence microscopy slides.
c) Table 3: Statistical significance has to be performed by Kruskal-Wallis and a post hoc-test.
d)Table 4: Some data are already described on table 1. The authors should show only the gender and BMI corrected data.
e) Regression analysis instead Table 4 could help to find the main independent associated factors with hypertrophic epicardial adipocytes.
e) Conclusion: Hypertrophic adipocytes size are associated with insulin resistance, low plasma circulating adiponectin levels and CAD severity.
Author Response
Dear reviewer!
The authors thank you for evaluating our article and for comments. We tried to correct the manuscript as much as possible according with your comments.
a) Table 1: EAT thickness and Gensini score should be included.
EAT thickness and Gensini score were included to the Table 1.
b) Figure 1: Control and Insulin treatment should be included in the fluorescence microscopy slides.
No clear visible differences were observed in the visual assessment of ROS under the influence of insulin under a microscope. Differences in the ROS production were recorded only with digital fluorimetry on the microplate rider and were about 11%, which was not visually distinguishable.
c) Table 3: Statistical significance has to be performed by Kruskal-Wallis and a post hoc-test.
Thank you very much for the error noticed in the caption to Table 3 - the initial comparison of several groups was carried out by the Kruskal-Wallis test. Now, in the latest version, we recalculated the statistics using ANOVA with Duncan's post hoc test. As you can see, some statistical differences were found compared to the Kruskal-Wallis test.
d) Table 4: Some data are already described on table 1. The authors should show only the gender and BMI corrected data.
We transferred duplicate and not-adjusted data from Table 4 to table 1. Now there are only the gender or BMI corrected data in the table 4.
e) Regression analysis instead Table 4 could help to find the main independent associated factors with hypertrophic epicardial adipocytes.
The applicability of classical regression models is limited by the requirements of the normality of the variables (all variables are only quantitative), the presence of a linear relationship between the dependent variable and the independent, the absence of a high correlation of the independent variables with each other, and the normality of the model residuals. Unfortunately, our variables don't have these properties.
When used as a binary variable, the presence of EAT hypertrophy а logistic regression of the dependence of EAT adipocyte hypertrophy on the severity of atherosclerosis, the level of adiponectin and indicators of carbohydrate metabolism was made.
EAT hypertrophy (yes/no) = -2.57 +11 FGluc – 6.7 FIns – 4.05 Adip# + 0.0334 GS*
where FGluc - fasting glucose, mM; Fins - fasting insulin, µIU/ml; Adip# - serum adiponectin, μg/ml, adjusted to gender and BMI; GS* - Gensini score, points, adjusted to gender.
Since the goal of our work was to study the pathophysiological mechanism of the formation of EAT hypertrophy and there was no goal to identify the diagnostic / prognostic significance of EAT hypertrophy, ROC analysis was not performed.
Therefore, at the moment we have used Spearman's nonparametric correlation methods as well.
e) Conclusion: Hypertrophic adipocytes size are associated with insulin resistance, low plasma circulating adiponectin levels and CAD severity.
We corrected conclusion.
At the end, I want to thank you again for your biased and thorough analysis of our article. Your remarks allowed me to discover some important points for further reflection on the problem under study and some mathematical approaches, the significance of which I underestimated.
Natalia V Naryzhnaya
Reviewer 2 Report
The authors addressed my previous comments in a satisfactory manner. I will only have one minor comment. 1. Abbreviations should be explained in the first place, separately in the abstract and next in the main text and under the tables / graphs. eg CAD in the introduction; Tab. 1 CAD, EAT, WHR…
Author Response
Dear reviewer!
The authors thank you for evaluating our article and for comments. We tried to correct the manuscript as much as possible according with your comments.
- Abbreviations should be explained in the first place, separately in the abstract and next in the main text and under the tables / graphs. eg CAD in the introduction; Tab. 1 CAD, EAT, WHR…
The decoding’s of abbreviations were added.
Natalia V Naryzhnaya
Reviewer 3 Report
All concerns are addressed.
Author Response
Dear reviewer!
The authors thank you for evaluating our article.
Natalia V Naryzhnaya
This manuscript is a resubmission of an earlier submission. The following is a list of the peer review reports and author responses from that submission.
Round 1
Reviewer 1 Report
The authors have tried to shed light on the association among epicardial adipocytes hypertrophy, insulin resistance and Gensini score.They have included blood and epicardial fat samples from 24 patients. After epicardial adipocytes isolation they have measured the diameter by microscopy. Then, insulin resistance was measured by ROS and glycerol determination. In addition the association among these parameters and Gensini score was performed. There are several major concerns that have to improved:
a) The authors did not split population regarding gender. This point is very important because the adiposity differs between women and men. The adipocyte size might be also different as well as the percentage of CAD presence.
b)In the same way, the population should be split regarding CAD or Gensini score.
c) Adipocyte size and hormones should be corrected by the associated factors (BMI, sex, CAD, etc...). This point is important
d) Percentage of low and high adipocytes sized on each patients should be represented.
e) Glucose uptake or consumption should be quantified on ex vivo assays.
f) An exhaustive explanation from where is taken epicardial fat sample
Author Response
Dear reviewer!
The authors thank you for evaluating our article and for comments. We tried to correct the manuscript as much as possible according with your comments.
a) The authors did not split population regarding gender. This point is very important because the adiposity differs between women and men. The adipocyte size might be also different as well as the percentage of CAD presence.
а) Indeed, atherosclerosis is closely related to gender, so we divided the population by gender and adjusted the scores by gender. It was revealed that in our sample of patients there are significant differences in leptin and the adiponectin / leptin index between men and women, as well as statistically insignificant, but multiple differences in the degree of atherosclerosis. At the same time, we did not reveal any differences in adiposity between men and women, the groups did not have differences in BMI, waist circumference, EAT thickness and EAT adipocyte size.
b) In the same way, the population should be split regarding CAD or Gensini score.
b) All patients included in study had CAD. Unfortunately, our small pilot study did not allow us to divide patients according to the severity of CAD.
c) Adipocyte size and hormones should be corrected by the associated factors (BMI, sex, CAD, etc...). This point is important
с) In our sample of patients, the dependence of leptin level on BMI was revealed, we adjusted this parameter by BMI. However, we did not consider it appropriate to adjust the size of the EAT adipocyte by BMI, since this would level the participation of obesity as a pathological growth factor of EAT in the formation of atherosclerosis, which was the goal of our study.
d) Percentage of low and high adipocytes sized on each patients should be represented.
d) We presented statistics on the number of large and small adipocytes in our sample patients (Table 2).
e) Glucose uptake or consumption should be quantified on ex vivo assays.
е) Glucose uptake ex vivo assays are very difficult methods for adult EAT adipocytes of elderly patients. Cytoplasm of this cells is very thin to quantified, for example, fluorescent glucose uptake. I attach fluorescent microscopy of glucose fluorescent analog 2-NBDG inclusion into the EAT adipocytes (300 nM 2-NBDG, 2 h, 37°C). It can be seen that 2-NBDG uptake is observed in single cells (see fig); parallel digital determination of glucose incorporation on a microplate reader does not give adequate results. An addition initial insulin resistance of EAT is very high. There is no articles about use ex vivo fluorescent assays of glucose uptake or consumption on EAT model, single publications about of method of insulin-stimulated radiolabeled glucose uptake [doi:10.1152/ajpendo.00384.2015]. But we have not the radio assay equipment and the permission to work with radioactive labels.
At the same time, insulin-induced ROS production has an advantages. Due to the fact that non-fluorescent DCP-DA must be dehydrogenated by intracellular enzymes to fluorescent DCP, it is all inside the cell and there is practically no extracellular fluorescence. This is an important point for digital detection on the reader, there is practically no background light (see fig in attached file). In addition, the DCF amplifies the ROS signal and can be successfully detected on a microplate reader. Since this is an indirect method for the determination of insulin resistance, we supported these data with results on insulin-dependent inhibition of lipolysis.
f) An exhaustive explanation from where is taken epicardial fat sample
f) Epicardial fat tissue explants was taken from the tissue surrounding the proximal parts of the right coronary artery. In all cases, electrocoagulation and other types of thermal and wave effects on tissues were not used for biopsy. We added this remark to the manuscript.

Reviewer 2 Report
The work of Naryzhnaya et al. is an interesting and valuable work worth considering for publication in the Biomedicines journal. However, it does contain a few small imperfections which I list below:
- Despite the use of the "list of abbreviations", each of them should be explained in the first place where they appear, separately in the abstract and the main text (e.g. in the "CAD" abstract).
- the number of the approval obtained from the local ethics committee should be indicated.
- BMI is not the most appropriate indicator of obesity, did the authors assess the content of adipose tissue, muscle mass, for example, through techniques with bioimpedance measurement?
- Line 138: what does it means “Warm Krebs-Ringer buffer was added” description is too general
- The methodological descriptions in the "Experimental section / Material and methods" section are too general and do not include the key terms of the analysis data. In my opinion, it should be expanded. e.g. in the part assessing the sensitivity of adipocytes to insulin, we do not know how many of them were per well, under what conditions the assessment was carried out whether on a laboratory table or in an incubator, whether equal percentages of live and dead cells were preserved, if not, the test may in fact assess the indicated parameter unreliably . What was the concentration of 2,3-dihydrodichlorofluorescein diacetate used? Lypolisis inhibition is also too general, apparatus was not indicated.
- In the descriptions of the graphs, the authors should avoid the indication group 1 / group 2 as it does not adequately characterize the given population. Charts and tables should follow the self-explanatory principle and have all abbreviations developed, without having to look for information in other parts of the work. This aspect needs to be improved.
- Allocation of patients to the group, and the allocation of groups should be described in the material and methods section. I suggest introducing a name other than group 1 and group 2. Table 1 should contain the characteristics not only in general, but also separately for groups 1 and 2, with an assessment of any statistically significant differences. Moreover, none of the cells in the table should be empty.
- Figure 3 has marks (a) and (b) in the wrong places and a unit [µm] is missing in the description of the Y axis. Figure 4 does not describe the Y axis at all, it is insufficient to provide the information in the description under the figure.
Author Response
Dear reviewer!
The authors thank you for evaluating our article and for comments. We tried to correct the manuscript as much as possible according with your comments.
1. Despite the use of the "list of abbreviations", each of them should be explained in the first place where they appear, separately in the abstract and the main text (e.g. in the "CAD" abstract).
1.the number of the approval obtained from the local ethics committee should be indicated.
1-2. Added the decoding of abbreviations, the number of the Ethics Commission approval protocol
3. BMI is not the most appropriate indicator of obesity, did the authors assess the content of adipose tissue, muscle mass, for example, through techniques with bioimpedance measurement?
3. Bioimpedance measurement were perform at the part of patient, but not give any additional advantages to our measurement and calculation.
4. Line 138: what does it means “Warm Krebs-Ringer buffer was added” description is too general
5. The methodological descriptions in the "Experimental section / Material and methods" section are too general and do not include the key terms of the analysis data. In my opinion, it should be expanded. e.g. in the part assessing the sensitivity of adipocytes to insulin, we do not know how many of them were per well, under what conditions the assessment was carried out whether on a laboratory table or in an incubator, whether equal percentages of live and dead cells were preserved, if not, the test may in fact assess the indicated parameter unreliably . What was the concentration of 2,3-dihydrodichlorofluorescein diacetate used? Lypolisis inhibition is also too general, apparatus was not indicated.
4-5. We have expanded and described the techniques in more detail.
6. In the descriptions of the graphs, the authors should avoid the indication group 1 / group 2 as it does not adequately characterize the given population. Charts and tables should follow the self-explanatory principle and have all abbreviations developed, without having to look for information in other parts of the work. This aspect needs to be improved.
6. We renamed the groups in the tables and introduced the decoding of the abbreviations in the graphs and tables.
7. Allocation of patients to the group, and the allocation of groups should be described in the material and methods section. I suggest introducing a name other than group 1 and group 2. Table 1 should contain the characteristics not only in general, but also separately for groups 1 and 2, with an assessment of any statistically significant differences. Moreover, none of the cells in the table should be empty.
7. We described in the Methods section the division of patients into groups and entered the clinical data for patients in groups 1 and 2 in Table 1.
8. Figure 3 has marks (a) and (b) in the wrong places and a unit [µm] is missing in the description of the Y axis. Figure 4 does not describe the Y axis at all, it is insufficient to provide the information in the description under the figure.
8. We change the figures description and abbreviatioin.

Reviewer 3 Report
Naryzhnaya et al. reported the association between the severity of coronary artery disease and the characters of EAT-derived adipocytes. The reviewer enjoyed the manuscript. To improve the quality of the manuscript, the reviewer requests the authors to address concerns below.
Major concerns:
- Figure 4 and Table 6: The authors should show the insulin resistance by the quantification of ROS generation against insulin (as the authors are doing in Figure 4), and the correlations between the ROS generation and other parameters (as in Table 6) in each group. The difference in insulin resistance between groups would be of the interest of the readers.
- The authors should add discussions about the future strategy to utilize the characterization of EAT-derived adipocytes for the prevention of coronary artery diseases. Would it be more advantageous than previously reported methods for the evaluation of metabolic syndrome?
Minor concerns:
- Abbreviations should be spelled out at the first appearance (e.g. "HOMA-IR" "DCF").
Author Response
Dear reviewer!
The authors thank you for evaluating our article and for comments. We tried to correct the manuscript as much as possible and answer on your questions.
1. Figure 4 and Table 6: The authors should show the insulin resistance by the quantification of ROS generation against insulin (as the authors are doing in Figure 4), and the correlations between the ROS generation and other parameters (as in Table 6) in each group. The difference in insulin resistance between groups would be of the interest of the readers.
Unfortunately, due to the small volume of the explant and the impossibility of carry out this study in all patients, it was impossible to assess the insulin-dependent ROS production and inhibition of lipolysis by groups depending on the size of the adipocyte. We added this remark to the manuscript.
2. The authors should add discussions about the future strategy to utilize the characterization of EAT-derived adipocytes for the prevention of coronary artery diseases. Would it be more advantageous than previously reported methods for the evaluation of metabolic syndrome?
Our study is one of the few studies in which the cellular mechanisms of the relationship of epicardial adipose tissue (EAT) with the severity of coronary atherosclerosis are studied in a clinical setting. However, our ultimate ambitious goal is to generate new important data to improve cardiovascular risk stratification in patients with metabolic syndrome.
We believe that for this we have to fulfill the following main tasks:
1) to continue our studies of intraoperative biopsy samples of EAT in the comparison group (in patients without coronary atherosclerosis who underwent surgery on the heart valves);
2) to establish non-invasive markers of hypertrophy and dysfunction of adipocytes EAT;
3) on a larger sample, make an attempt to find gender differences in the relationship between hypertrophy / hyperplasia of adipocytes EVT with the severity of coronary atherosclerosis;
4) to plan prospective observation of the patients included in the study with an assessment of their long-term prognosis and patency of coronary artery bypass grafts.
